# A comparative study of dengue virus vectors in major parks and adjacent residential areas in Ho Chi Minh City, Vietnam

**Trang Thi Thuy Huynh**[1,2,3]**, Noboru Minakawa**[2]*

**1** Department of Medical Entomology and Zoonotics, Pasteur Institute in Ho Chi Minh City, Ho Chi Minh City, Vietnam, **2** Institute of Tropical Medicine (NEKKEN), Nagasaki University, Nagasaki, Japan, **3** Graduate School of Biomedical Sciences, Nagasaki University, Nagasaki, Japan

* minakawa@nagasaki-u.ac.jp

**Data Availability Statement:** All relevant data are within the manuscript and its Supporting Information files.

## Abstract

The primary dengue virus vectors, *Aedes aegypti* and *Aedes albopictus*, are primarily daytime biting mosquitoes. The risk of infection is suspected to be considerable in urban parks due to visitor traffic. Despite the importance of vector control for reducing dengue transmission, little information is available on vector populations in urban parks. The present study characterized mosquito habitats and estimated vector densities in the major urban parks in Ho Chi Minh City, Vietnam and compared them with those in adjacent residential areas. The prevalences of habitats where *Aedes* larvae were found were 43% and 9% for the parks and residential areas, respectively. The difference was statistically significant (prevalence ratio [PR]: 5.00, 95% CI: 3.85–6.49). The prevalences of positive larval habitats were significantly greater in the parks for both species than the residential areas (PR: 1.52, 95% CI: 1.04–2.22 for *A. aegypti*, PR: 10.10, 95% CI: 7.23–14.12 for *A. albopictus*). Larvae of both species were positively associated with discarded containers and planters. *Aedes albopictus* larvae were negatively associated with indoor habitats, but positively associated with vegetation shade. The adult density of *A. aegypti* was significantly less in the parks compared with the residential areas (rate ratio [RR]; 0.09, 95% CI: 0.05–0.16), while the density of *A. albopictus* was significantly higher in the parks (RR: 9.99, 95% CI: 6.85–14.59). When the species were combined, the density was significantly higher in the parks (RR: 2.50, 95% CI: 1.92–3.25). The urban parks provide suitable environment for *Aedes* mosquitoes, and *A. albopictus* in particular. Virus vectors are abundant in the urban parks, and the current vector control programs need to have greater consideration of urban parks.

## Author summary

The primary dengue virus vectors, *Aedes aegypti* and *Aedes albopictus*, are primarily daytime biting mosquitoes and therefore the risk of infection may be considerable in urban parks due to human foot traffic. Prior to the present study little information was available on vector populations in urban parks. Here we describe that larvae of both species were positively associated with discarded containers and planters. *Aedes albopictus* larvae were

**Funding:** This research (NM) was funded by the Japan Agency for Medical Research and Development (AMED) under Grant Number JP21wm0125006, https://www.amed.go.jp/koubo/01/06/0106C_00018.html. The funders had no role in study design, data collection and analysis, decision to publish, or preparation of the manuscript.

**Competing interests:** The authors have declared that no competing interests exist.

negatively associated with indoor habitats, but positively associated with vegetation shade. *Aedes albopictus* was predominant in the urban parks while *A. aegypti* was predominant in adjacent residential areas. When the species were combined the density of vectors was greater in the urban parks. The current vector control programs need to take into consideration vector intensity within urban parks.

## Introduction

According to the WHO, over the last two decades 100 to 400 million dengue virus infections occurred annually, and the number of cases increased eight-fold [1]. Dengue is primarily an urban disease, and the recent increase is likely associated with the rapid expansion of urban areas [2]. Between the years 2000 and 2015 over 70% of outbreaks occurred in urban areas or across a combination of urban and rural areas [3]. The dense human populations of urban areas increase vector-human contact, and dengue virus (DENV) circulates more efficiently [2]. However, recent seroprevalence studies showed that rural populations have been exposed to DENV as much as urban populations [4,5]. The high seropositive rate in rural areas is likely due to an increase of mobile populations infected with DENV from urban areas, and transitions of rural areas toward urban areas.

The principal DENV vector species, *Aedes aegypti* and *Aedes albopictus*, are well adapted to urban environments. Larvae of both species are often found in discarded containers in residential areas with poor sanitation [6–9]. This is one of the reasons why high infection risk is often associated with socio-economic factors and rapid urbanization without proper infrastructure [2]. As a result, the abundance of vectors increases with urban sprawl in tropical regions. While the former species breeds almost exclusively in small anthropogenic habitats in urban areas [10], the latter species may also breed in tree holes and cut or broken bamboos [11].

Feeding activities of both vector species are primarily diurnal [12,13]. Although they often display a bimodal peak activity during the morning and late afternoon [14–16], the peak hours may vary depending on environmental conditions [17]. Because people may spend much of the day away from home, the risk of infection is not limited to residential areas. A study in Ho Chi Minh City, Vietnam, reported that residents spent more hours at home were less likely IgM positive with DENV [18]. In the case of a dengue outbreak in central Tokyo in 2014, nearly all infected persons had visited the same parks during the epidemic [19], and the populations of *A. albopictus* in the parks had extremely high biting densities and infection rates [20,21]. Numerous discarded containers are usually present in urban parks, and vegetation provides a suitable environment for *A. albopictus* [22,23]. Although the scale of the outbreak in Tokyo was much smaller compared with epidemics in tropical areas, the incident implies that urban parks may also facilitate intense transmission in endemic regions, because many people of all age classes visit parks during the daytime when the vectors are active.

Although *A. albopictus* is the only dengue virus vector in Japan, both *A. aegypti* and *A. albopictus* are present in tropical and sub-tropical Asia. While *A. albopictus* is associated more with vegetation or rural areas [22–25], *A. aegypti* is predominant in large cities such as Ho Chi Minh City, Singapore, and Yangon [26–29]. Despite the potential risk of infection in urban parks, little information is available for the vector populations in urban parks except for a study in São Paulo, Brazil [11,30].

The present study addressed four hypotheses comparing the populations of *A. albopictus* and *A. aegypti* in the major parks and the adjacent residential areas in Ho Chi Minh City.

Specifically: 1) the relative abundance of *A. albopictus* adults is greater in the parks compared with *A. aegypti* because of the greater vegetation cover, 2) more natural larval breeding habitats are available in the parks, 3) the preference of breeding habitat types is different between the two mosquito species, and 4) the abundance of both vector species in the parks is comparable to that in the residential areas. Evaluating these hypotheses provides important information for future vector control strategies.

## Methods

### Study site

Ho Chi Minh City is the most populous city in Vietnam. The population in 2019 was nearly 9 million (over 8% of the total population of the country) within 2,061 km$^2$, and nearly 7 million in the 19 inner districts (urban area: 283 km$^2$) where we conducted the present study (Fig 1) [31]. The city had approximately 210 parks in 2014, and the present study focused on the six largest and most popular parks (Gia Dinh, Le Thi Rieng, Phu Lam, Tao Dan, Thao Cam Vien,

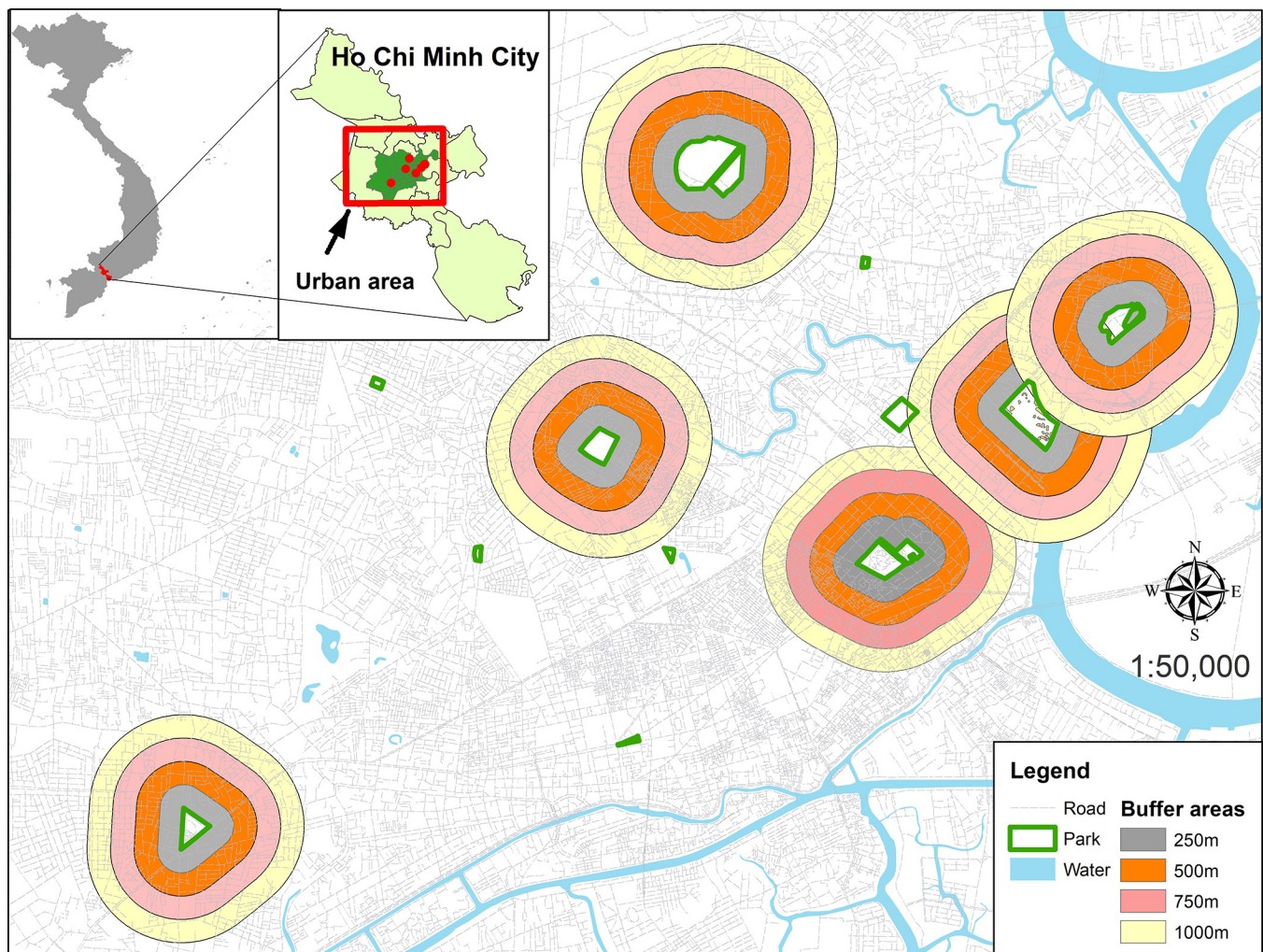

**Fig 1. The study sites in Ho Chi Minh City.** The six major parks and their adjacent residential areas in the central part (urban area) of the city. The distances of 250 m, 500 m, 750 m and 1,000 m from the park boundary were shown in the residential areas. The base map was adapted from an open source map retrieved on Natural Earth Data at http://www.naturalearthdata.com/about/terms-of-use/.

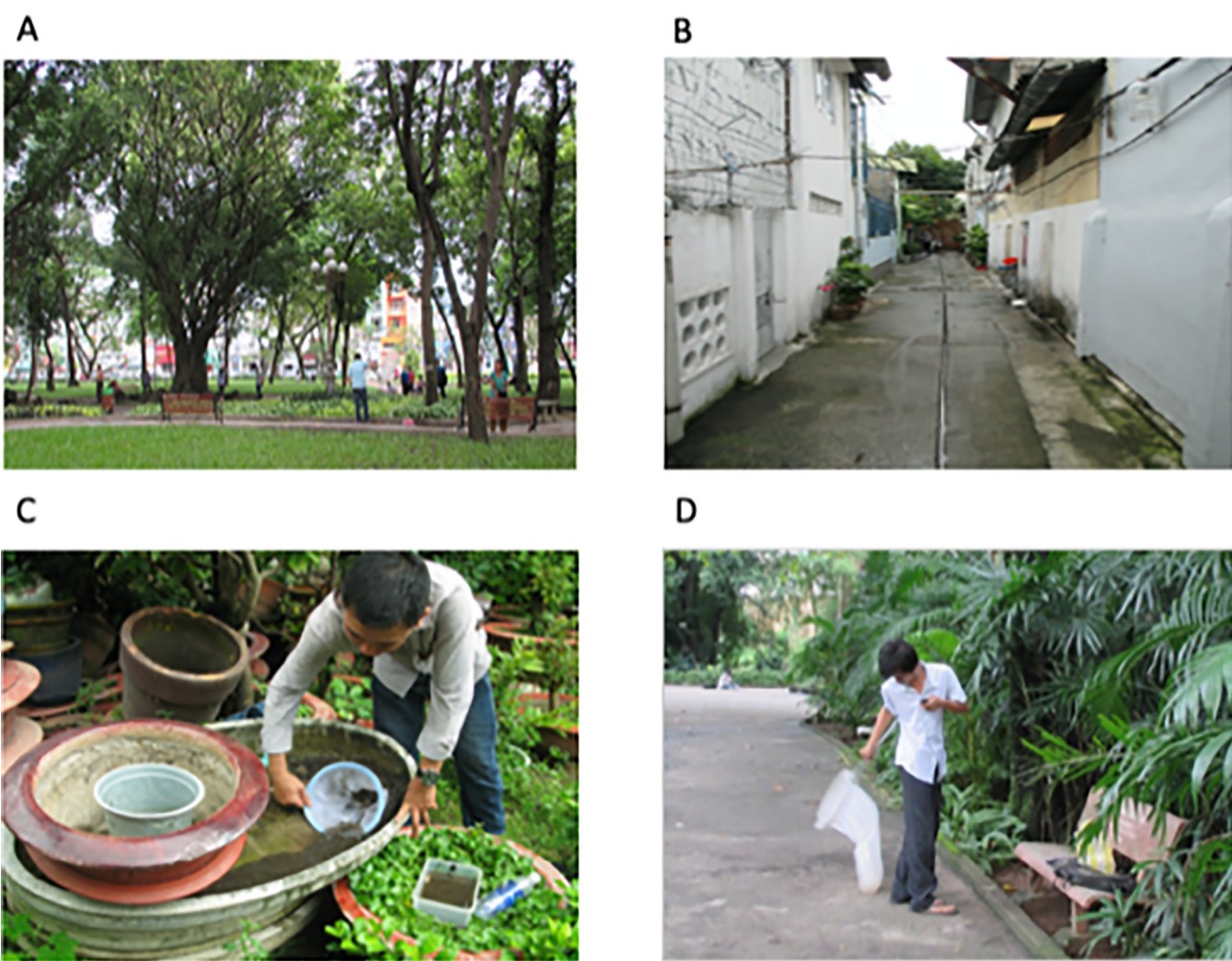

**Fig 2. Photographs taken in the study area.** (A) park, (B) residential area, (C) sampling larvae from planters in a courtyard, and (D) sampling adults in one of the parks.

and Van Thanh) in the urban area and their adjacent residential areas for comparison (Fig 2A and 2B). The median size of the parks is 8.9 ha (IQR = 7.7). These parks are located in the central part of the city. The parks have several scattered tall trees. The understory is mainly covered with short grass, and there are several gardens and footpaths. Thao Cam Vien has a zoo, and Le Thi Rieng, Phu Lam andVan Thanh have a pond. Reportedly, 2.4 million people visited Thao Cam Vien in 2014 [32].

High dengue virus transmission mainly occurs during the rainy season in the second half of the year [22]. Dengue morbidity and mortality increased in recent years. After 6,715 dengue cases were reported in the city in 2014 [33], the number reached 68,540 in 2019 [34]. The local dengue control programs mainly employ reactive insecticide spraying around houses of reported cases [18]. However, this was not the case in the study sites before and during the present study in the 2014 season. The control programs do not use larvicides in the city. A study conducted between 2010 and 2013 reported that the prevalence of DENV infection in *A. aegypti* was 1% in the residential area in the central part of the city [18]. An amino-acid substitution, F1534C, in the voltage-sensitive sodium channel related to knockdown resistance (*kdr*)

has been reported from the *A. albopictus* population in Ho Chi Minh City [35]. A high frequency of another point mutation, F1269C, related to *kdr* has been also reported from the *A. aegypti* population in the city [36].

## Sampling *Aedes* larvae

*Aedes* mosquito larvae were collected throughout the parks during the period between September and early October 2014, in the middle of the rainy season. Larvae were obtained from potential water holding habitats with pipets, dippers (white, 350ml, 13 cm in diameter, 91.5 cm handle, BioQuip Products, Rancho Dominquez, USA), or aquatic nets (white, 20 cm in diameter, 33 cm in depth, 100 μm mesh, the standard net used in the National Dengue Control Programme of Vietnam) (Fig 2C) [37]. We searched for larval habitats not only outdoor, but also inside of buildings such as office building and shed in the parks. Habitat types were classified into nine categories: bucket, cup, discarded containers, plant, planter, jar, water tank, flower vase, and other. Habitat locations were categorized into indoor, vegetation shade, under eaves, and open outdoor. The larval survey excluded ponds, ditches, puddles, and running water—locations where *A. aegypti* and *A. albopictus* are unlikely to inhabit. When habitats were small enough, field workers tried to capture all larvae using aquatic nets and dippers. For larger habitats, an aquatic net was passed through the water at least five times [37]. Collected larvae were preserved in 75% ethanol and identified to species in the laboratory. Three to eight field workers were involved in the larval survey for each park.

In an adjacent residential area, 20 transect lines were established from the boundary of each park using the geographical information system, GIS (ArcGIS ESRI Inc, Redlands, CA, USA) (S1 Fig). The transect lines were separated from each other by roughly equal distances. For the larval survey the four closest houses were selected 250 m from the park boundary on arbitrarily selected 1 of 20 lines. The next sampling site was established at 500 m from the park on the next line clockwise. Similarly, two sampling sites were established at distances of 750 m and 1000 m on the next two lines, respectively. This process was repeated with the remaining transect lines, and a total of 80 houses (4 houses x 20 transect lines) were selected for each adjacent residential area. A total of 480 houses (80 houses x 6 areas) were targeted for six adjacent residential areas. We adapted this systematic design to minimize traveling time between sites. We excluded the areas that were overlapped with the residential areas adjacent to the other parks. Field workers searched for potential water holding habitats throughout the indoor and outdoor habitats. Habitat types and their locations were classified into the categories described for the parks. In a small habitat, all larvae were captured with aquatic nets and dippers. For larger habitats, an aquatic net was passed through the water at least five times. Collected larvae were preserved in 75% ethanol and identified to species. Three to seven field workers were involved in the larval survey for each residential area.

## Sampling *Aedes* adults

The adult mosquito survey was conducted during the same period as the larval survey. Within each park the areas where visitors cannot access were identified through field observation, and the inaccessible areas were excluded from the survey. In the accessible areas, 48 equally spaced sampling sites were established using GIS (S2A Fig). From 48 sites, 12 equally spaced sites were selected for sampling adult mosquitoes between 6:00 and 8:00 on the first day (S2B Fig). At each site, we tried to establish six equally spaced sampling points in an area of 100 $m^2$ (12 sites x 6 sampling points = 72 sampling points for the morning sampling); however, the space between sampling points was often arbitrary because of presence of obstacles such as a tree (S2C Fig). A collector stood at each sampling point, and for 1 min collected approached

mosquitoes with a sweep net (30 cm diameter white insect net, BioQuip, Rancho Dominguez, California, USA) followed by a pause of 1 min (Fig 2D). We chose the net collection method over human landing catches because it reduces risk of virus infection, but still captures host seeking mosquitoes. The cycle of sampling and pausing was repeated five times at each point, for a total of 5 min of sampling (a total of 10 min including pauses). Captured mosquitoes were transferred to a 15 ml plastic centrifuge tube using an aspirator during the interval pause. The sampling procedure was repeated at the five remaining points. Twelve collectors participated in the survey to complete samplings at 12 sites in the morning hours. In the afternoon hours (between 16:00 and 18:00), this process was repeated at 12 different sites, and at 24 remaining sites in the same park on the following day (two days for each park). Since *A. albopictus* and *A. aegypti* often display a bimodal peak activity [14–16], we sampled adults during the morning hours and the afternoon hours. The adult survey was planned to have a total of 288 samplings for each park and a total of 1,728 sampling points within the six parks.

The adult mosquito collection was conducted in each adjacent residential area on the same day the survey was conducted in the corresponding park. In each residential area, equally spaced 16 lines were drawn from the park boundary using GIS (S3A Fig). We reduced the number of the transect lines, because the adult survey required more time for each house than the larval survey. The lines were systematically assigned four different distances in the same manner for the larval survey. At each distance site, the three closest houses were identified. Within each house, three equally spaced sampling points were established in the yard and in the house, respectively, for a total of six sampling points (S3B Fig). The survey targeted four different distance lines in the morning hours and in the afternoon hours on the first day, respectively. Mosquitoes were sampled at the remaining eight lines on the following day. The survey planned to have a total of 288 sampling points with 48 houses in each residential area, and a total of 1,728 sampling points for six residential areas. A total of 3,456 sampling points was assigned for the parks and residential areas. Eight field workers collected mosquitoes in each residential area.

## Statistical analysis

Correspondence analysis with the R package "MASS" was used to explore the relationships between larval habitat types, habitat locations, and distances from the parks [38]. The relationships of these variables with each mosquito species were also examined with correspondence analysis. The occurrence of mosquito species at each potential larval habitat was categorized into "presence of both species", "*A. aegypti* only", "*A. albopictus* only", and "absence of both species." A logistic regression model was used to compare the larval occurrences of each species between the habitat locations, between indoor and outdoor habitats, between each distance from the parks and parks, and between the parks and residential areas using the R-package "lme4" and "glmmTMB" [39,40]. Habitat types, geographical areas, habitat locations, and distances were considered as potential covariates. A geographical area was defined as an area including a park and its adjacent residential area. Independence of each covariate was assessed with variation inflation factor, chi-square test, and Pearson product moment correlation or Cramér's V with R package "recompanion" [41]. Collectors, dates, and sampling sites were considered as potential random factors, and a caterpillar plot was used to determine whether these variables should be included in the model. Since log-binomial mixed models with the frequentist approaches often fail to converge, and the outcomes were binary, a prevalence ratio (PR) and confidence interval were estimated with Poisson regression models [42,43].

Adult densities between the parks and residential areas were compared with quasi-Poisson regression models, Poisson regression models, negative binomial models, or zero-inflated models; and collectors, dates, and sampling sites were considered as potential random factors. An adult density was defined as the number of mosquitoes per sampling point (sampled for 5 min). The geographical areas and sampling time (morning or afternoon) were considered as potential confounding factors. Since adult mosquitoes were sampled from indoor and outdoor habitats at each house, the densities of each mosquito species were also compared between both habitats.

## Results

### Occurrence of larvae

The larval survey found 1,988 potential habitats that were holding water. Buckets were most abundant followed by flower vases, cups, and discarded containers (Table 1). A correspondence analysis showed that cups, buckets, vases, and water tanks were associated with the residential areas regardless of distance from the parks, while discarded containers, plants, and planters were associated with the parks (Fig 3A). Cups, buckets, and vases were associated with indoor habitats; and jars, planters, and water tanks were associated with roof eaves and open outdoor habitats in the residential areas (Fig 3B). Discarded containers were associated with vegetation shade (Fig 3B), and most vegetation shade habitats were found in the parks (Table 2 and Fig 3C). The difference in the composition of potential habitat types between the residential areas and parks was statistically significant (Table 3).

*Aedes* larvae were found at 295 (20%) of 1,988 habitats. Discarded containers and planters composed 60% of the positive habitats. Despite large numbers of buckets and cups, the occurrence of *Aedes* larvae were significantly lower in these habitats than the other habitats while their presence was significantly higher in discarded containers than the other habitats (Table 4). At a species level, *A. aegypti* and *A. albopictus* were found at 9% and 16% of the potential habitats, respectively. A correspondence analysis revealed that both species were positively associated with discarded containers and planters, and *A. albopictus* was negatively associated with flower vases, buckets, water tanks, jars, and cups (Fig 3D). The occurrences of both species were significantly lower in cups than the other habitats. While the occurrence of *A. albopictus* was also significantly lower in water tanks compared with the other habitat types,

**Table 1. Numbers (%) of water holding potential larval habitat types found in the parks, the residential areas, four different distances from the parks, three different outdoor locations and indoor location.**

| | Bucket | Cup | Discard | Jar | Plant | Planter | Tank | Vase | Uncategorized | Total |
|---|---|---|---|---|---|---|---|---|---|---|
| Park | 23 (3.6) | 16 (2.5) | 143 (22.2) | 45 (7.0) | 37 (5.8) | 144 (22.4) | 47 (7.3) | 19 (3.0) | 169 (26.3) | 643 |
| Residential | 406 (30.2) | 246 (18.3) | 48 (3.6) | 73 (5.4) | 2 (0.2) | 33 (2.5) | 136 (10.1) | 250 (18.6) | 151 (11.2) | 1345 |
| 250 m | 80 (18.4) | 53 (20.2) | 13 (6.8) | 26 (22.0) | 1 (2.6) | 9 (5.1) | 33 (18.0) | 74 (27.5) | 37 (11.6) | 326 |
| 500 m | 117 (27.3) | 86 (32.8) | 11 (5.8) | 13 (11.0) | 1 (2.6) | 6 (3.4) | 38 (20.8) | 59 (22.0) | 44 (13.8) | 375 |
| 750 m | 105 (24.5) | 43 (16.4) | 8 (4.2) | 19 (16.1) | 0 | 12 (6.8) | 29 (15.9) | 19 (16.1) | 21 (6.6) | 302 |
| 1000 m | 104 (24.2) | 64 (24.4) | 16 (8.4) | 15 (12.7) | 0 | 6 (3.4) | 36 (19.7) | 15 (12.7) | 49 (15.3) | 342 |
| Indoor | 332 (34.4) | 226 (23.4) | 8 (0.8) | 35 (3.6) | 0 (0) | 10 (1.0) | 76 (7.9) | 207 (21.4) | 72 (7.5) | 966 |
| Outdoor | 97 (9.5) | 36 (3.5) | 183 (17.9) | 83 (8.1) | 39 (3.8) | 167 (3.8) | 107 (10.5) | 62 (6.1) | 28 (24.3) | 1022 |
| Under eaves | 52 (23.9) | 16 (7.4) | 26 (11.9) | 17 (7.8) | 0 (0) | 13 (6.0) | 23 (10.6) | 23 (10.6) | 48 (22.0) | 218 |
| Vegetation shade | 20 (5.5) | 2 (0.6) | 102 (28.2) | 15 (4.1) | 28 (7.7) | 74 (20.4) | 15 (4.1) | 4 (1.1) | 102 (28.2) | 362 |
| Open outdoor sites | 25 (5.7) | 18 (4.1) | 55 (12.4) | 51 (11.5) | 11 (2.5) | 80 (18.1) | 69 (15.6) | 35 (7.9) | 98 (22.2) | 442 |
| Total | 429 (21.6) | 262 (13.2) | 191 (9.6) | 118 (5.9) | 39 (2.0) | 177 (8.9) | 183 (9.2) | 269 (13.5) | 320 (16.1) | 1988 |

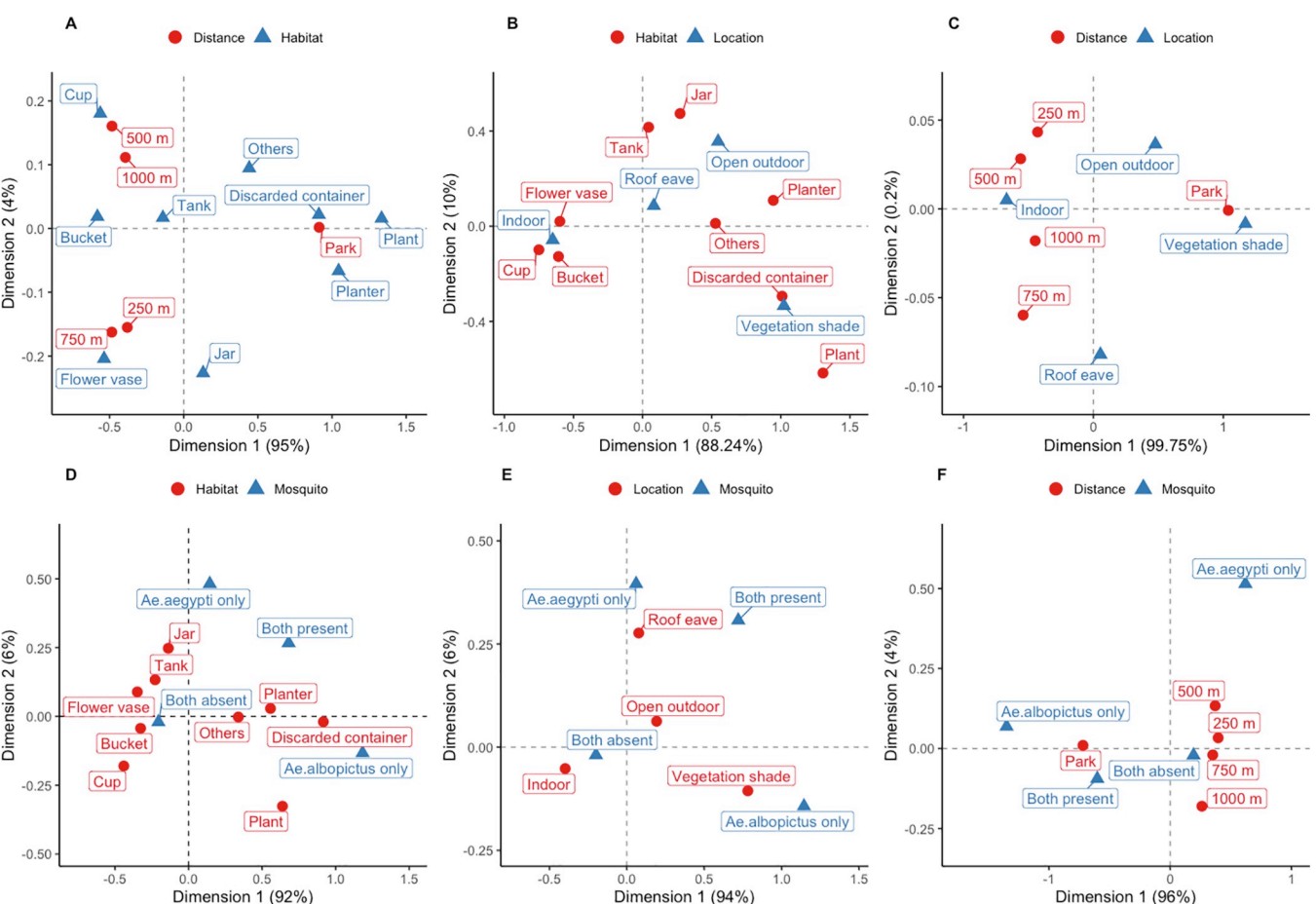

**Fig 3. Correspondence analysis plots.** (A) Between habitat types and distances from the parks, (B) habitat types and locations, (C) habitat locations and distances, (D) habitat types and mosquitoes, (E) habitat locations and mosquitoes, and (F) distances and mosquitoes.

it was significantly higher in discarded containers. The logistic regression analysis for each habitat type was adjusted for the variables of distance, habitat location, and geographical area; but the variable of residential/park was excluded because of the high association with habitat types (Table 3). The optimal models included only sampling site as a random intercept.

While the occurrence of *A. aegypti* was positively associated with indoor habitat, roof eaves, and open outdoor habitats, *A. albopictus* was positively associated with vegetation shade (Fig 3E). The occurrence was significantly higher with all outdoor habitats for both species

**Table 2. Numbers (%) of indoor and outdoor water holding potential larval habitats found in the parks and four different distances from the parks.**

|  | **Indoor** | **Under eaves** | **Vegetation shade** | **Open outdoor** | **All outdoor** | **Total** |
|---|---|---|---|---|---|---|
| Park | 11 (1.7) | 76 (11.8) | 315 (49.0) | 241 (37.5) | 632 (98.3) | 643 |
| Residential | 955 (71.0) | 142 (10.6) | 47 (3.5) | 201 (14.9) | 393 (29.0) | 1348 |
| 250 m | 221 (67.8) | 32 (14.7) | 15 (4.6) | 58 (17.8) | 105 (32.2) | 326 |
| 500 m | 280 (74.7) | 35 (9.3) | 8 (2.1) | 52 (13.9) | 95 (25.3) | 375 |
| 750 m | 219 (72.5) | 37 (12.3) | 7 (2.3) | 39 (12.9) | 83 (27,5) | 302 |
| 1000 m | 235 (68.7) | 38 (11.1) | 17 (5.0) | 52 (15.2) | 107 (31.3) | 342 |
| Total | 966 (48.6) | 218 (11.0) | 362 (18.2) | 442 (22.2) | 1022 (51.4) | 1988 |

**Table 3. Correlations between the variables associated with larval habitats.**

|  | Chi-square | df | p-value | Cramér V |
|---|---|---|---|---|
| Residential/Park versus geographical area | 102 | 5 | < 0.001 | 0.23 |
| Residential/Park versus distance from the parks | 1988 | 4 | < 0.001 | 1 |
| Residential/Park versus habitat | 792 | 8 | < 0.001 | 0.63 |
| Residential/Park versus habitat location | 1024 | 3 | < 0.001 | 0.71 |
| Geographical area versus distance from the parks | 175 | 20 | < 0.001 | 0.16 |
| Geographical area versus habitat | 237 | 40 | < 0.001 | 0.16 |
| Geographical area versus habitat location | 145 | 15 | < 0.001 | 0.16 |
| Habitat versus distance from the parks | 837 | 32 | < 0.001 | 0.33 |
| Habitat location versus distance from the parks | 1031 | 12 | < 0.001 | 0.42 |
| Habitat versus habitat location | 1043 | 24 | < 0.001 | 0.42 |

compared with indoor habitats (Table 5). The variable of residential/park was excluded from these regression models because its association was high with habitat locations (Table 3). The optimal models included only sampling site as a random intercept.

The proportions of habitats containing *Aedes* larvae were 44% and 9% for the parks and residential areas, respectively. Logistic regression analyses revealed that the proportion was

**Table 4. Numbers (%) of positive sites where larvae of each mosquito species were found in each water holding potential habitat type and prevalence ratio (PR).**

|  | Bucket | Cup | Discard | Jar | Plant | Planter | Tank | Vase | Uncategorized | Total |
|---|---|---|---|---|---|---|---|---|---|---|
| *Ae. aegypti* |  |  |  |  |  |  |  |  |  |  |
| Positive (%) | 20 (4.7) | 0 (0) | 33 (17.3) | 16 (13.6) | 2 (5.1) | 28 (15.8) | 19 (10.4) | 20 (7.4) | 38 (11.9) | 176 (8.9) |
| Total | 429 | 262 | 191 | 118 | 39 | 177 | 183 | 269 | 320 | 1988 |
| Ref Positive (%) | 156 (10.0) | 176 (10.2) | 143 (9.0) | 160 (8.6) | 174 (8.9) | 148 (8.2) | 157 (8.7) | 156 (9.1) | 138 (8.3) | - |
| Total | 1559 | 1726 | 1797 | 1870 | 1949 | 1811 | 1805 | 1719 | 1668 |  |
| Unadjusted PR | 0.46* | 0 | 1.93* | 1.51 | 0.60 | 1.91* | 1.07 | 0.93 | 1.31 | - |
| 95% CI | 0.28–0.78 | 0 | 1.25–2.97 | 0.86–2.68 | 0.14–2.56 | 1.19–3.09 | 0.64–1.79 | 0.56–1.55 | 0.88–1.96 |  |
| Adjusted PR | 0.61 | 0* | 1.51 | 1.37 | 0.50 | 1.49 | 0.95 | 1.36 | 1.00 | - |
| 95% CI | 0.37–1.00 | 0 | 0.98–2.32 | 0.78–2.38 | 0.12–2.15 | 0.93–2.39 | 0.57–1.57 | 0.81–2.29 | 0.67–1.48 |  |
| *Ae. albopictus* |  |  |  |  |  |  |  |  |  |  |
| Positive (%) | 20 (4.7) | 1 (0.4) | 93 (48.7) | 13 (11.0) | 14 (35.9) | 66 (37.3) | 16 (8.7) | 9 (3.3) | 89 (27.8) | 321 (16.1) |
| Total | 429 | 262 | 191 | 118 | 39 | 177 | 183 | 269 | 320 | 1988 |
| Ref Positive (%) | 301 (19.3) | 320 (18.5) | 228 (12.7) | 308 (16.5) | 307 (15.4) | 225 (14.1) | 305 (16.9) | 312 (18.2) | 232 (13.9) | - |
| Total | 1559 | 1726 | 1797 | 1870 | 1949 | 1811 | 1805 | 1719 | 1668 |  |
| Unadjusted PR | 0.33* | 0.03* | 2.73* | 0.63 | 2.26* | 1.90* | 0.55* | 0.31* | 1.65* | - |
| 95% CI | 0.20–0.54 | 0–0.26 | 1.95–3.80 | 0.34–1.18 | 1.04–4.90 | 1.26–2.86 | 0.32–0.95 | 0.15–0.63 | 1.19–2.28 |  |
| Adjusted PR | 0.86 | 0.08* | 1.74* | 0.64 | 0.93 | 1.01 | 0.56* | 0.75 | 1,10 | - |
| 95% CI | 0.52–1.40 | 0.01–0.59 | 1.32–2.29 | 0.35–1.16 | 0.51–1.70 | 0.73–1.39 | 0.33–0.94 | 0.37–1.52 | 0.84–1.44 |  |
| Both species |  |  |  |  |  |  |  |  |  |  |
| Positive (%) | 28 (6.5) | 40 (15.3) | 102 (53.4) | 23 (19.5) | 15 (38.5) | 75 (42.4) | 24 (13.1) | 23 (8.6) | 103 (32.2) | 295 (19.9) |
| Total | 429 | 262 | 191 | 118 | 39 | 177 | 183 | 269 | 320 | 1988 |
| Ref Positive (%) | 367 (23.5) | 394 (22.8) | 292 (16.5) | 372 (19.9) | 380 (19.5) | 320 (17.7) | 371 (20.6) | 372 (21.6) | 292 (17.5) | - |
| Total | 1559 | 1726 | 1797 | 1870 | 1949 | 1811 | 1805 | 1719 | 1668 |  |
| Unadjusted PR | 0.32* | 0.02* | 2.64* | 0.94 | 2.04* | 1.82* | 0.63* | 0.54* | 1.65* | - |
| 95% CI | 0.21–0.48 | 0–0.17 | 1.98–3.52 | 0.59–1.51 | 1.03–4.06 | 1.29–2.59 | 0.40–0.99 | 0.34–0.88 | 1.25–2.18 |  |
| Adjusted PR | 0.65* | 0.05* | 1.66* | 0.91 | 0.96 | 1.07 | 0.66 | 1.15 | 1.12 | - |
| 95% CI | 0.43–0.98 | 0–0.35 | 1.28–2.14 | 0.58–1.43 | 0.54–1.70 | 0.80–1.44 | 0.42–1.01 | 0.72–1.83 | 0.87–1.44 |  |

A PR (95% CI) was estimated considering the habitats other than the subjected habitat as a reference (PR = 1), and an asterisk shows statistical significance with a p-value of 0.05.

**Table 5. Numbers (%) of positive sites where larvae of each mosquito species were found in indoor and outdoor locations and prevalence ratio (PR).**

|  | Indoor | Under eaves | Vegetation shade | Open outdoor | All outdoor sites | Total |
|---|---|---|---|---|---|---|
| *Ae. aegypti* |  |  |  |  |  |  |
| Positive | 34 (3.5) | 37 (17.0) | 50 (13.8) | 55 (12.4) | 142 (13.9) | 176 (8.9) |
| Negative | 932 (96.5) | 181 (83.0) | 312 (86.2) | 387 (87.6) | 880 (86.1) | 1812 (91.1) |
| Unadjusted PR | 1 (ref) | 4.80 (2.89–7.99)* | 3.62 (2.25–5.82)* | 3.44 (2.17–5.44)* | 3.79 (2.54–5.65)* | - |
| Adjusted PR | 1 (ref) | 3.79 (2.23–6.40)* | 2.89 (1.60–5.24)* | 2.54 (1.52–4.26)* | 3.03 (1.90–4.85)* | - |
| *Ae. albopictus* |  |  |  |  |  |  |
| Positive | 16 (1.7) | 43 (19.7) | 159 (43.9) | 103 (23.3) | 305 (29.8) | 321 (16.1) |
| Negative | 950 (98.4) | 175 (80.3) | 203 (56.1) | 339 (76.7) | 717 (70.2) | 1667 (83.9) |
| Unadjusted PR | 1 (ref) | 11.93 (6.56–21.67)* | 22.94 (13.39–39.33)* | 12.36 (7.15–21.38)* | 15.44 (9.15–26.05)* | - |
| Adjusted PR | 1 (ref) | 4.22 (2.20–8.08)* | 4.07 (2.11–7.88)* | 3.45 (1.81–6.56)* | 3.83 (2.07–7.11)* | - |
| Both species |  |  |  |  |  |  |
| Positive | 38 (3.9) | 60 (27.5) | 174 (48.1) | 123 (27.8) | 357 (34.9) | 395 (19.9) |
| Negative | 928 (96.1) | 158 (72.5) | 188 (51.9) | 319 (72.2) | 665 (65.1) | 1593 (80.1) |
| Unadjusted PR | 1 (ref) | 7.5 (4.60–10.81)* | 11.10 (7.64–16.14)* | 6.44 (4.39–9.45)* | 8.06 (5.66–11.46)* | - |
| Adjusted PR | 1 (ref) | 3.66 (2.30–5.80)* | 3.42 (2.12–5.52)* | 2.72 (1.73–4.29)* | 3.16 (2.06–4.84)* | - |
| Total | 966 | 218 | 362 | 442 | 1022 | 1988 |

A PR (95% CI) was estimated considering the locations other than the subjected location as a reference (PR = 1), and an asterisk shows statistical significance with a p-value of 0.05.

significantly higher in the parks for both species (Table 6). However, a correspondence analysis indicated that *A. albopictus* was positively associated with the parks while *A. aegypti* was positively associated with the residential areas (Fig 3F). Although the relationships of *A. albopictus* occurrence were not clear with the distances from the parks (Fig 3F), the occurrence of *A. aegypti* was not significantly lower at the distances of 500 m and 750 m than the parks (Table 6). In particular, the PR at 500 m was as high as that of the parks. Nevertheless, the

**Table 6. Numbers (%) of positive sites where larvae of each mosquito species were found at four distances from the parks and prevalence ratio (PR).**

|  | 0 m (Park) | 250 m | 500 m | 750 m | 1000 m | All residential sites | Total |
|---|---|---|---|---|---|---|---|
| *Ae. aegypti* |  |  |  |  |  |  |  |
| Positive | 73 (11.4) | 19 (5.8) | 38 (10.1) | 23 (7.6) | 23 (6.7) | 103 (7.7) | 176 (8.9) |
| Negative | 570 (88.6) | 307 (94.2) | 337 (89.9) | 279 (92.4) | 319 (93.3) | 1242 (92.3) | 1812 (91.1) |
| Unadjusted PR | 1 (ref) | 0.52 (0.28–0.95)* | 0.95 (0.58–1.58) | 0.73 (0.41–1.29) | 0.57 (0.3–1.01) | 0.69 (0.47–1.01) | - |
| Adjusted PR | 1 (ref) | 0.51 (0.28–0.94)* | 0.95 (0.58–1.55) | 0.63 (0.36–1.11) | 0.54 (0.30–0.95)* | 0.66 (0.45–0.96)* | - |
| *Ae. albopictus* |  |  |  |  |  |  |  |
| Positive | 265 (41.2) | 5 (1.5) | 15 (4.0) | 12 (4.0) | 24 (7.0) | 56 (4.2) | 321 (16.1) |
| Negative | 378 (58.8) | 321 (98.5) | 360 (96.0) | 290 (96.0) | 318 (93.0) | 1289 (95.8) | 1667 (83.9) |
| Unadjusted PR | 1 (ref) | 0.04 (0.02–0.10)* | 0.10 (0.06–0.17)* | 0.10 (0.05–0.18)* | 0.18 (0.11–0.28)* | 0.10 (0.07–0.14)* | - |
| Adjusted PR | 1 (ref) | 0.04 (0.02–0.10)* | 0.10 (0.06–0.17)* | 0.09 (0.05–0.16)* | 0.17 (0.11–0.27)* | 0.10 (0.07–0.14)* | - |
| Both species. |  |  |  |  |  |  |  |
| Positive | 279 (43.4) | 22 (6.7) | 40 (10.7) | 27 (8.9) | 27 (7.9) | 116 (8.6) | 395 (19.9) |
| Negative | 364 (56.6) | 304 (93.3) | 335 (89.3) | 275 (91.1) | 315 (92.1) | 1229 (91.4) | 1593 (80.1) |
| Unadjusted PR | 1 (ref) | 0.16 (0.10–0.26)* | 0.26 (0.18–0.38)* | 0.22 (0.14–0.34)* | 0.19 (0.12–0.29)* | 0.21 (0.16–0.17)* | - |
| Adjusted PR | 1 (ref) | 0.16 (0.10–0.26)* | 0.26 (0.16–0.38)* | 0.20 (0.13–0.31)* | 0.18 (0.12–0.28)* | 0.20 (0.15–0.26)* | - |
| Total | 643 | 326 | 375 | 302 | 342 | 1345 | 1988 |

A PR (95% CI) was estimated considering a distance of 0 m (park) as a reference (PR = 1), and an asterisk shows statistical significance with a p-value of 0.05.

reduction in proportion of positive sites was similar among all distances when both species were combined. In the regression models for comparing parks and residential areas, the covariates included were only residential/park and geographical area, because the associations of residential/park were high with the other variables (Table 3). The optimal models also included sampling site as a random intercept, but not other variables.

## Adult mosquito abundance

Adult mosquito sampling was conducted at 2,640 (76%) of the 3,456 initially planned sampling points, and the missed sites were largely due to bad weather conditions. Of the sampled sites, 24 were excluded from the analyses because of incomplete datasets. For the resulting 2,616 points, 1,584 (61%) and 1,032 (39%) were in the parks and residential areas, respectively (Table 7). A total of 1,117 *Aedes* mosquitoes were collected, 185 (17%) of which were *A. aegypti* and 932 (83%) were *A. albopictus*.

The numbers of mosquitoes collected in the parks and the residential areas were 897 (80%) and 220 (20%), respectively. The numbers of *A. aegypti* and *A. albopictus* were 24 (3%) and 873 (97%) in the parks, and 161 (73%) and 61 (27%) in the residential areas, respectively. The density of *A. aegypti* was significantly greater in the residential areas than the parks, and it was more than ten-fold (Table 7). The density of *A. aegypti* was greater at all distances in the residential areas compared with the parks, and the density was greatest at 250 m. The increases of *A. aegypti* were statistically significant at all distances from the parks. In contrast, the density of *A. albopictus* was significantly higher in the parks than the residential areas. The adjusted rate rations (RRs) indicate that the reductions of *A. albopictus* were around 90% in the residential areas, and the reductions were greater at 750 m and 1000 m compared with the parks.

**Table 7. Comparisons of adult mosquito density (no / sampling point) between the parks and four different distances from the parks in the residential areas.**

|  | 0 m (Park) | 250 m | 500 m | 750 m | 1000 m | All residential sites |
|---|---|---|---|---|---|---|
| *Ae. aegypti* |  |  |  |  |  |  |
| Average (No / point) | 0.02 | 0.25 | 0.09 | 0.17 | 0.11 | 0.16 |
| Median (range) | 0 (0–3) | 0 (0–4) | 0 (0–2) | 0 (0–5) | 0 (0–2) | 0 (0–5) |
| Unadjusted RR (95% CI) | 1 (ref) | 15.63* (7.49–32.61) | 8.23* (3.54–19.10) | 11.70* (5.52–24.82) | 9.11* (4.22–19.69) | 11.31* (6.25–20.46) |
| Adjusted RR (95% CI) | 1 (ref) | 15.12* (7.29–31.37) | 8.29* (3.60–19.08) | 11.90* (5.65–25.06) | 9.36* (4.35–20.12) | 11.35* (6.29–20.50) |
| *Ae. albopictus* |  |  |  |  |  |  |
| Average (No / point) | 0.55 | 0.08 | 0.08 | 0.02 | 0.05 | 0.06 |
| Median (range) | 0 (0–9) | 0 (0–3) | 0 (0–2) | 0 (0–1) | 0 (0–3) | 0 (0–3) |
| Unadjusted RR (95% CI) | 1 (ref) | 0.13* (0.07–0.25) | 0.15* (0.08–0.30) | 0.04* (0.02–0.11) | 0.08* (0.04–0.16) | 0.10* (0.06–0.14) |
| Adjusted RR (95% CI) | 1 (ref) | 0.14* (0.08–0.26) | 0.16* (0.08–0.30) | 0.04* (0.02–0.11) | 0.08* (0.04–0.17) | 0.10* (0.07–0.15) |
| Both species |  |  |  |  |  |  |
| Average (No / point) | 0.57 | 0.33 | 0.17 | 0.19 | 0.16 | 0.21 |
| Median (range) | 0 (0–9) | 0 (0–4) | 0 (0–2) | 0 (0–5) | 0 (0–2) | 0 (0–5) |
| Unadjusted RR (95% CI) | 1 (ref) | 0.57* (0.37–0.87) | 0.38* (0.23–0.63) | 0.34* (0.21–0.54) | 0.30* (0.18–0.48) | 0.39* (0.30–0.50) |
| Adjusted RR (95% CI) | 1 (ref) | 0.58* (0.39–0.88) | 0.38* (0.23–0.63) | 0.35* (0.22–0.56) | 0.31* (0.19–0.49) | 0.40* (0.31–0.52) |
| Total sampling points | 1,584 | 270 | 216 | 270 | 276 | 1,032 |

A rate ratio (RR: 95% CI) for each distance was estimated based on the mosquito density in the parks, and an asterisk shows statistical significance with a p-value of 0.05.

**Table 8. Comparisons of adult mosquito abundance between the indoor and outdoor locations, and between the morning and afternoon hours.**

| | Indoor | Outdoor | Morning | Afternoon |
|---|---|---|---|---|
| *Ae. aegypti* | | | | |
| Average (No / point) | 0.21 | 0.10 | 0.15 | 0.16 |
| Median (range) | 0 (0–4) | 0 (0–4) | 0 (0–5) | 0 (0–4) |
| Unadjusted RR (95% CI) | 1 (ref) | 0.46* (0.33–0.65) | 1 (ref) | 0.90 (0.52–1.56) |
| Adjusted RR (95% CI) | 1 (ref) | 0.47* (0.30–0.73) | 1 (ref) | 0.82 (0.49–1.37) |
| *Ae. albopictus* | | | | |
| Average (No / point) | 0.06 | 0.06 | 0.06 | 0.05 |
| Median (range) | 0 (0–3) | 0 (0–3) | 0 (0–3) | 0 (0–3) |
| Unadjusted RR (95% CI) | 1 (ref) | 0.97 (0.59–1.60) | 1 (ref) | 1.20 (0.84–1.70) |
| Adjusted RR (95% CI) | 1 (ref) | 1.03 (0.62–1.72) | 1 (ref) | 1.38* (1.04–1.84) |
| Both species | | | | |
| Average (No / point) | 0.27 | 0.16 | 0.22 | 0.21 |
| Median (range) | 0 (0–5) | 0 (0–4) | 0 (0–5) | 0 (0–4) |
| Unadjusted RR (95% CI) | 1 (ref) | 0.57* (0.42–0.76) | 1 (ref) | 1.14 (0.87–1.49) |
| Adjusted RR (95% CI) | 1 (ref) | 0.58* (0.43–0.77) | 1 (ref) | 1.24 (0.98–1.59) |
| Total sampling points | 528 | 528 | 1,475 | 1,141 |

A rate ratio (RR: 95% CI) for outdoor locations was estimated based on indoor location density, and a RR (95% CI) for the afternoon hours was estimated based on the morning hour density. An asterisk shows statistical significance with a p-value of 0.05.

Overall, the densities of *Aedes* mosquitoes in the parks were significantly greater compared with the residential areas, and reduced with an increase of distance from the parks. The adult data were analyzed with quasi-Poisson regression models that incorporated sampling site as a random factor. The variables of geographical area and sampling period were included as confounding factors.

Quasi-Poisson models revealed that the density of *A. aegypti* in the residential areas was significantly less in the outdoor habitats compared with the indoor habitats (Table 8). However, the difference in density between the habitats was not statistically significant for *A. albopictus*. When the two species were combined, the density in the residential areas was significantly lower in the outdoor habitats than the indoor habitats. The optimal regression models also included geographical area and sampling period as confounding factors and sampling site as a random factor.

Between the morning and afternoon sampling hours, the difference in density was not statistically significant for *A. aegypti*; however, the density of *A. albopictus* was significantly greater in the afternoon (Table 8). When the two species were combined, the difference in density between the morning and afternoon hours was not statistically significant. The optimal quasi-Poisson models included geographical area and park/residential as confounding factors and sampling site as a random factor.

## Discussion

The present study found that the densities of adult *Aedes* mosquitoes, in particular *A. albopictus*, were greater in the major parks in Ho Chi Minh City than the adjacent residential areas.

While nearly all *Aedes* mosquitoes collected in the parks were *A. albopictus*, *A. aegypti* was predominant in the adjacent residential areas. Past studies reported a positive association of *A. albopictus* with vegetation in rural areas [22–24]. The present study confirmed a similar phenomenon within an urban area at a smaller spatial scale. The study in São Paulo also found a positive association of *A. albopictus* with vegetation in a much larger urban park which includes a remaining natural forest [44]. The vegetation in the parks of Ho Chi Minh City is apparently sparser than in the park of São Paulo, but is sufficient to provide a suitable environment for this mosquito species.

Most larval breeding habitats in the parks were not found directly on plants; rather, in artificial habitats such as discarded containers and planters in vegetation shade. The results still suggest that shade created by abundant vegetation in the parks provides a suitable environment for *A. albopictus*. Nutrients from leaf litter may become an energy source and enhance the development of larvae [45]. *Aedes albopictus* adults are known to ingest sugars from understory vegetation [46,47], and they may seek blood meals more actively in the shade created by trees [48].

For both mosquito species the prevalence of positive larval habitats was lower in the residential areas than in the parks, suggesting that the residential environment is less suitable regardless of the taxa. One plausible explanation is that the residential habitats are unstable. Residents physically remove potential habitats more often and replace or empty the water in the habitats; and as a result the habitat lifespans in the residential areas become shorter. Vector control activities may also explain the lower prevalence of larvae, and the lower density of adults in the residential areas. The local government mainly employ reactive insecticide spraying around houses of reported cases [18]; however, this was not the case for the houses which we sampled from before and during the present study in the 2014 season. Residents may still use insecticides for personal protection, but the effects of individual effort on mosquitoes may not be extensive.

The present study found that the density of *A. aegypti* was higher in the residential areas than in the parks; and in particular the density was greater inside houses. This mosquito species is considered to be the most efficient vector of DENV because of its greater anthropophagy and virus susceptibility compared with *A. albopictus* [49,50]. Although the overall density of *A. aegypti* was lower than that of *A. albopictus* in the present study, the contact between human and this mosquito species should be greater in houses, and DENV may be efficiently transmitted.

The density of *A. aegypti* was greatest at 250 m distance from the parks, and the density of *A. albopictus* was higher at 250 m and 500 m compared with the other distances. These phenomena may be explained by a spillover of mosquitoes from the parks to the close residential areas. Otherwise, mosquitoes may be concentrated in the area to seek blood meals from both residents and park visitors. While the former case may be applied to *A. albopictus* which is abundant in the parks, the latter case may be applied to *A. aegypti* which is predominant in the residential areas. It is known that the oviposition of *A. aegypti* is positively associated with the peripheral area of the urban park in São Paulo [11,30].

The abundance of *A. albopictus* in the parks may extend the transmission season. As the dry season progresses, the abundance of *A. albopictus* increases relative to *A. aegypti* [51], because the latter species produces superior desiccation-resistant eggs to survive the dry season [52,53]. Vegetation in the parks may retain humidity and provide nutrients for *A. albopictus* during the dry season. As a result, viral transmission may persist in the parks during the dry season via the *A. albopictus* population.

Moreover, urban parks may increase the risk of other viral diseases such as chikungunya and Zika providing habitats to *A. albopictus*. An evidence of chikungunya virus infection has

been reported from Ho Chi Minh City [54], and most Zika cases in the 2016–2017 outbreak in Vietnam were reported from the city [55]. Reportedly, *A. albopictus* is able to transmit the chikungunya virus variant with the E1-226V mutation more efficiently than *A. aegypti* [56]. The E1-226V mutation was first identified during the chikungunya outbreak in the African Indian Ocean islands in 2005 [57], and the variant has been already introduced into China, India, Malaysia, Thailand and Taiwan [58–62]. The variant can be introduced to Vietnam in near future.

## Limitation

The present study was conducted cross-sectionally in the middle of the rainy season, and temporally and spatially only covered a small portion of the vector population. Therefore, the information from the preset study is too little to understand the entire vector population dynamics in the parks and the adjacent residential areas in the city. Although a large number of larvae were sampled systematically, the data from the present study were qualitative. Because the sampling methods differed according to types and sizes of larval habitat, it is not easy to standardize larval densities among different habitats [37].

## Implication

The role of urban parks in viral transmission should not be underestimated, and urban parks should be included in current vector control programs. As suggested by past studies [6–9], the present study confirms the importance of artificial vessels such as discarded containers for breeding mosquitoes in the urban parks. The results suggest that periodic removal of discarded containers in the parks is important for vector control. Managing over-grown understory vegetation not only reduces the quality of larval habitats [45], but also adult mosquito resting sites [63]. The study results regarding the distances from the parks suggest that the importance of the peripheral area for vector control; however, further studies are needed to confirm this notion.

Although the comparison of vector populations among the parks was not an aim of the present study, the information should be useful for the local vector control programs. The results from the present study imply that the difference in vegetation cover may reflect on the difference in distribution and abundance of vectors among the parks. Identifying the factors related to the distribution of breeding habitats in the parks will also become valuable information for vector control.

## Supporting information

**S1 Fig. Sampling design for mosquito larvae in the residential areas**
(TIFF)

**S2 Fig. Sampling design for adult mosquitos in the parks.** (A) 48 sampling sites in each park, (B) 12 sites for each of four sampling periods in two days, and (C) six sampling points at each sapling site.
(TIFF)

**S3 Fig. Sampling design for adult mosquitoes in the residential areas.** (A) three closest houses at each of 16 sampling sites at four different distances from the parks, and (B) six sampling points within each house.
(TIFF)

**S1 Dataset. Dataset for larave.**
(CSV)

**S2 Dataset. Dataset for adults.**
(CSV)

## Acknowledgments

The first author is grateful for the academic support from the Program for Nurturing Global Leaders in Tropical and Emerging Communicable Diseases, Graduate School of Biomedical Sciences, Nagasaki University. We thank the staff of Pasteur Institute in Ho Chi Minh City (PIHCMC) for laboratory and field assistance as well as the staff from Preventive Medicine Center in Ho Chi Minh City (HCDC) and from the districts and communes. In addition, we thank Mark L. Wilson, Takashi Tsunoda and Ataru Tsuzuki for constructive comments.

## Author Contributions

**Conceptualization:** Trang Thi Thuy Huynh, Noboru Minakawa.

**Data curation:** Trang Thi Thuy Huynh.

**Formal analysis:** Noboru Minakawa.

**Funding acquisition:** Noboru Minakawa.

**Investigation:** Trang Thi Thuy Huynh.

**Methodology:** Trang Thi Thuy Huynh, Noboru Minakawa.

**Project administration:** Trang Thi Thuy Huynh, Noboru Minakawa.

**Resources:** Trang Thi Thuy Huynh, Noboru Minakawa.

**Supervision:** Trang Thi Thuy Huynh, Noboru Minakawa.

**Validation:** Trang Thi Thuy Huynh.

**Writing – original draft:** Trang Thi Thuy Huynh, Noboru Minakawa.

**Writing – review & editing:** Trang Thi Thuy Huynh, Noboru Minakawa.

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
