## [Decision Letter · Decision Letter 0]

9 Sep 2021

Dear Dr. Minakawa,

Thank you very much for submitting your manuscript "A comparative study of dengue virus vectors in major parks and adjacent residential areas in Ho Chi Minh City, Vietnam" for consideration at PLOS Neglected Tropical Diseases. As with all papers reviewed by the journal, your manuscript was reviewed by members of the editorial board and by several independent reviewers. In light of the reviews (below this email), we would like to invite the resubmission of a significantly-revised version that takes into account the reviewers' comments. 

We cannot make any decision about publication until we have seen the revised manuscript and your response to the reviewers' comments. Your revised manuscript is also likely to be sent to reviewers for further evaluation.

Sincerely,

Pedro F. C. Vasconcelos

Deputy Editor

Pedro Vasconcelos

Deputy Editor

Reviewer's Responses to Questions

**Key Review Criteria Required for Acceptance?**

**Methods**

-Are the objectives of the study clearly articulated with a clear testable hypothesis stated?

-Is the study design appropriate to address the stated objectives?

-Is the population clearly described and appropriate for the hypothesis being tested?

-Is the sample size sufficient to ensure adequate power to address the hypothesis being tested?

-Were correct statistical analysis used to support conclusions?

-Are there concerns about ethical or regulatory requirements being met?

Reviewer #1: Study site:

- Please add the vector control (both larva and adult) situation in the study sites and what actions are currently treat in the area including the resistant report (if available).

- More describe about the geographical area, people demographic and occupation that present the potential of human-vector contact.

- Good systemic design for the protocol of larva collection but the too complicate to understand overall. 

Line 150: 

- what method did you use for adult collection? Please define clearly. Do you use the swipe method for 5 mins each point? Do you have any citation by this? Where location focus to collect adults? Rewrite this section.

- Please describe how many total sampling numbers and sites per park (228) (from total 6 parks).

- It would be better if the map of collection sites in the supplementary data; residence area and parks (with adjacent areas) are added that will image the distribution of collection.

Analysis part is good to support the conclusion and answer the objectives.

Reviewer #2: The methods was OK but need to be more clearly explained to assure the reader got the idea. Check the comment and question in the Methods section.

The sample size, although not been large, but i think enough to draw the conclusion, since the results has clear pattern (at least for the overall results of the 6 parks)

Reviewer #3: Some limitations with the study design and sample size. See overall comments below.

**Results**

-Does the analysis presented match the analysis plan?

-Are the results clearly and completely presented?

-Are the figures (Tables, Images) of sufficient quality for clarity?

Reviewer #1: Results

- Good represent of the correlation results in Figure 1 among multi couple factors (A—F). 

- The results could image and understand the correlation among factors and mosquito abundance properly. 

- It would be nice indicator to show percent infection rate (or the previous study) in the obtaining mosquitoes and subsequently more impact to point the risk of transmission area.

- Line 288: what is the rationale of the sampling period related to the mosquito abundance? And how to related to the transmission risk unless done by the human landing rate?

Reviewer #2: The results, mostly, are as expected that A. albopictus was more vegetation dependent (park, outdoor) and A. aegypti was human-associated dependent (indoor, artificial containers). 

The results presented were based on the total results from all the six parks analyzed in the city, however it will be more clear and more interesting if the authors can show the one or two most important results of each park (not only the overall results after mixing the data from 6 parks) to see whether the pattern discussed are more or less in all the parks. It will be interesting to see how the pattern of each park on the preference of park vs residential area, or indoor vs outdoor for both A. aegypti and Ae. albopictus.

The boxplot graph in figure 2 show very few information, mostly similar between the species comparison. Even, can draw contradictory conclusion between the figure and the tables. If all the information covered by the tables presented, then better to remove figure 2.

In tables presented there were writing inconsistency for the value of references, some were '1' and the others were '1 ref' or '1 (ref)'.

Please also respond to the comments in the Results section.

Reviewer #3: Results are challenging to read as the many comparisons are not always clearly articulated. See additional detail below.

**Conclusions**

-Are the conclusions supported by the data presented?

-Are the limitations of analysis clearly described?

-Do the authors discuss how these data can be helpful to advance our understanding of the topic under study?

-Is public health relevance addressed?

Reviewer #1: Discussion

- Generally, many researches showed the non-correlation of vector abundance and transmission risks. How do you explain by this issue?

- Line 337-340: “The movements of people and viruses would be less within the residential areas compared with the parks, unless large commercial activities are present; and on the whole viruses would be transmitted more efficiently and effectively in the urban parks than the residential areas.” It there any evidence to show by this sentence? This discussion part is not related to the study. Otherwise, cut this section.

- Please more discussion for Ae. agypti that highly contact with human and what should be more focused in term of factor correlate-transmission risk.

Implication 

- Please highlight: what is the outstanding knowledge that obtained from this study and imply for vector control program in the area; focus location point of vector control priority, distance of insecticide spray, how to reduce the human-vector contact risk? And how to treat in the vegetable are/parks related to albopictus habitat?

- How the current larvicidal control operate in the area? That should be adjust to match with this study outcome?

Reviewer #2: The conclusion about the preference of sites and habitats are OK. However, the dengue transmission risk should not only considered by the abundance of vector species and its site habitat preferences, but also the abundance of host that they may encounter (host population in residential area are clearly higher than in park since the site is their home, in the morning peoples are still in their houses befor work, while afternoon most people are comeback to their houses. Although some may visit parks, but it may only less than 30% of population. It is good if authors can show data about density of host in parks vs residential area, so the conclusion will have bases. Also, the natural presence of endo symbiont such as Wolbachia spp. in A. albopictus but not in A. aegypti, that may affect the capacity vectorial of A. aegypti several folds higher than A. albopictus. 

Overall conclusion was OK, but need to add some discussion on the mentioned aspects.

Reviewer #3: Tend to jump off point of what this data shows. see specifics below.

**Editorial and Data Presentation Modifications?**

Reviewer #1: In the method part need to add the supplementary data of mosquito collection points as distribution pattern.

Reviewer #2: The Methods section should be write more clearly to not confuse readers whether one methods used for both park or residential, or only true for either park or residential area. The authors please check the comments in the pdf manuscript in method section and clarify according to the question.

Reviewer #3: (No Response)

**Summary and General Comments**

Reviewer #1: This is the good study to represent the correlation of multi factors and Dengue vectors abundance. The methodology design is good but need to be re-write for more clear and understanding. Analysis pattern is good and can be imaged for the potential impact to vectorial factors. Discussion need to add more particularly in the of the implication of the study outcome and the vector control program application. Only minor revision need to be completed.

Reviewer #2: This study is interesting, and give more insight to common readers, as well as officers who performed the vector control programs, emphasize of common knowledges about preference site and habitats of A. aegypti and A. albopictus with clear data. It is worth to published with a minor revision in method section (to be clearer) and additional discussion on the risk of viral transmission, especially dengue.

Reviewer #3: The authors present a qualitative mosquito survey of mosquitoes in park areas and nearby residential areas to assess the presence of two DENV vectors. There are many limitations of the study design and the paper is challenging to read but some interesting points did come out. Specific points for the authors to consider are noted below.

General: 

probably should define “urban parks” very early in the manuscript. This could be very different from playground style with groomed landscape to simply native plants in an otherwise Metropolitan/concrete focus. Are there large trees or mostly flowers? Is it basketball courts or open grassy areas? Providing some “visual” imagery of what type of urban park you are referring to would assist the readers in understanding how this might look different from a residential area (sub-urban presumably).

An additional general comment is the difficulty in reading the text as so many comparisons were made but it wasn’t always clear which items were being compared, presented, or discussed. Please consider reviewing the manuscript (throughout) to make certain each statement such as “urban area has more prevalence” is clear regarding what is “more” in the comparison.

Specific:

Line 52: aren’t albos dusk/dawn biters over general day biting? Please provide references for each species as a daytime biter.

Line 60: is there currently no vector control in parks? If some, what type are you referring to here? (ie. truck based spraying or trash removal?, etc.)

Line 77: not clear how transitions toward urban areas results in more seropositivity in rural areas. Might consider revising this sentence for clarity.

Line 109: are more natural egg-laying sites important when you’ve stated that these mosquitoes readily lay eggs in discarded trash? Also, when referring to “breeding sites”, do you really mean “egg-laying sites” (which is really the correct term even though breeding site is colloquially used). Might consider using the proper term.

Lines 121-124: might consider describing the ecology of the parks under study.

Lines 180-181: if a site had a single target mosquito on a single collecting trip, would this site have been considered positive? Another site could have 100 target mosquitoes on a single trip and yet would be considered “positive” as the first site? Seems the plus / minus approach might not indicate significant differences among the sites, correct? How does the study design account for numeric differences?

Figure 1 – the light coloring of the text in each graph makes them difficult to read.

Lines 228 and near – please use caution when describing something as having a higher or lower presence as sometimes, it is not clear if you mean higher in that container versus other container types or higher in that container type for one species versus the other species. Please make sure each comparison is clear.

Table 4 – text too small to read

Line 303-304- please provide a reference for this statement

Line 330-331 -- this line is a bit misleading as it doesn’t truly assess the time this species bites the most. It just compares one specific morning time versus one specific afternoon time. Since other times were not considered (ie – dusk), the sentence should be clarified.

Line 343 – this is a bit confusing as in the results, it was noted that both species were preferentially found in the outdoor sites. Please clarify.

Line 345-346 – this sentence seems out of place in this paragraph.

Line 372 – this statement is only partially accurate. Albopictus is only more susceptible to ECSA strains of CHIKV with a specific mutation at position 226. Other strains of CHIKV are typically better vectored by aegypti. Please correct this sentence.

Lines 373-375 – this sentence seems unnecessary here. Suggest deleting it

Line 389-90. – this sentence, while likely a good idea, is not based on anything presented in this paper. Suggest deleting it.

PLOS authors have the option to publish the peer review history of their article (what does this mean?). If published, this will include your full peer review and any attached files.

Reviewer #1: No

Reviewer #2: Yes: Isra Wahid

Reviewer #3: No
---

## [Decision Letter · Decision Letter 1]

21 Dec 2021

Dear Dr. Minakawa,

We are pleased to inform you that your manuscript 'A comparative study of dengue virus vectors in major parks and adjacent residential areas in Ho Chi Minh City, Vietnam' has been provisionally accepted for publication in PLOS Neglected Tropical Diseases.

Best regards,

Pedro F. C. Vasconcelos

Deputy Editor

Pedro Vasconcelos

Deputy Editor

Reviewer's Responses to Questions

**Key Review Criteria Required for Acceptance?**

**Methods**

-Are the objectives of the study clearly articulated with a clear testable hypothesis stated?

-Is the study design appropriate to address the stated objectives?

-Is the population clearly described and appropriate for the hypothesis being tested?

-Is the sample size sufficient to ensure adequate power to address the hypothesis being tested?

-Were correct statistical analysis used to support conclusions?

-Are there concerns about ethical or regulatory requirements being met?

Reviewer #1: The adjusted method part is acceptable. It answers the hypothesis and objectives with the suitable design and sample size. Data analysis is valuable to answer the multiple factors of location, containers, species, etc. Statistic analysis is acceptable. Figures of study design is better understanding and can be guideline for actual application.

Reviewer #2: (No Response)

Reviewer #3: (No Response)

**Results**

-Does the analysis presented match the analysis plan?

-Are the results clearly and completely presented?

-Are the figures (Tables, Images) of sufficient quality for clarity?

Reviewer #1: Results part are acceptable.

Reviewer #2: (No Response)

Reviewer #3: (No Response)

**Conclusions**

-Are the conclusions supported by the data presented?

-Are the limitations of analysis clearly described?

-Do the authors discuss how these data can be helpful to advance our understanding of the topic under study?

-Is public health relevance addressed?

Reviewer #1: The conclusion is verify. The study can shed the key container of Aedes mosquito breeding places. This can be used for analysis of the vector surveillance monitoring in the local area.

Reviewer #2: (No Response)

Reviewer #3: (No Response)

**Editorial and Data Presentation Modifications?**

Reviewer #1: The manuscript is acceptable.

Reviewer #2: (No Response)

Reviewer #3: (No Response)

**Summary and General Comments**

Reviewer #1: This study is valuable in term of the good guideline and study design of vector surveillance for Dengue control and monitoring. Vector abundance is the one of important key factor impact to the dengue transmission particularly in seasonal dynamic and breeding sties.

Reviewer #2: All reviewer comments and questions have been addressed by authors with additional explanation, figure and data. The manuscript have been re-written and much improved after authors follow the reviewer recommendation.

Reviewer #3: The authors have substantially updated the manuscript to address most of the suggestions of the reviewers. A few points still remain concerning - such as not changing the term "breeding site" to "oviposition site". While the authors are correct that many publications use the term "breeding site" - this is technically not correct and it would be good to see entomological leaders move to correct this issue. However, overall, the manuscript is substantially improved and does provide some useful information to the field.

PLOS authors have the option to publish the peer review history of their article (what does this mean?). If published, this will include your full peer review and any attached files.

Reviewer #1: No

Reviewer #2: **Yes: **Isra Wahid

Reviewer #3: No

---

## [Editor Report · Acceptance letter]

7 Jan 2022

Dear Dr. Minakawa,

We are delighted to inform you that your manuscript, "A comparative study of dengue virus vectors in major parks and adjacent residential areas in Ho Chi Minh City, Vietnam," has been formally accepted for publication in PLOS Neglected Tropical Diseases.

Best regards,

Shaden Kamhawi

co-Editor-in-Chief

Paul Brindley

co-Editor-in-Chief
